# Identification of efficient prokaryotic cell-penetrating peptides with applications in bacterial biotechnology

Hyang-Mi Lee [iD] [1], Jun Ren[1], Kha Mong Tran[1], Byeong-Min Jeon[2], Won-Ung Park[2], Hyunjoo Kim[1], Kyung Eun Lee[3], Yuna Oh[3], Myungback Choi[1], Dae-Sung Kim[2] & Dokyun Na[1]✉

In bacterial biotechnology, instead of producing functional proteins from plasmids, it is often necessary to deliver functional proteins directly into live cells for genetic manipulation or physiological modification. We constructed a library of cell-penetrating peptides (CPPs) capable of delivering protein cargo into bacteria and developed an efficient delivery method for CPP-conjugated proteins. We screened the library for highly efficient CPPs with no significant cytotoxicity in *Escherichia coli* and developed a model for predicting the penetration efficiency of a query peptide, enabling the design of new and efficient CPPs. As a proof-of-concept, we used the CPPs for plasmid curing in *E. coli* and marker gene excision in *Methylomonas* sp. DH-1. In summary, we demonstrated the utility of CPPs in bacterial engineering. The use of CPPs would facilitate bacterial biotechnology such as genetic engineering, synthetic biology, metabolic engineering, and physiology studies.

[1] Department of Biomedical Engineering, Chung-Ang University, Seoul, Republic of Korea. [2] Department of Biotechnology, Korea University, Seoul, Republic of Korea. [3] Advanced Analysis Center, Korea Institute of Science and Technology, Seoul, Republic of Korea. ✉email: blisszen@cau.ac.kr

In the field of biotechnology, protein delivery systems have various applications like laboratory applications (gene editing) and clinical applications (drug delivery)[1–7]. To date, numerous methods have been introduced for efficient intracellular protein delivery in mammalian cells, in which various biomaterials are used as delivery vehicles, including cationic lipids[2], peptides[3,8,9], nanoparticles[10–13], and decorated liposomes[14,15]. In mammalian cells, for in vivo applications, delivery systems should have properties such as cell-type-specific targeting[1,3,16], non-toxicity[5,17], and highly efficient cell uptake and release[4,18,19]. Furthermore, to improve protein delivery efficiency, various methods are also combined[13,20], and delivery vehicles are designed to be controlled by micro-environmental aspects (pH[14,15,21], light[16,22], or enzymes[23]).

Cell-penetrating peptides (CPPs) are generally composed of 10–20 amino acids and are derived primarily from protein-transduction domains. A variety of CPPs has been developed to transport various cargo (proteins, DNA, siRNAs, and other small molecules) into mammalian cells for diverse purposes[8,9,17,24–28]. For example, due to their low molecular weight and high permeability, CPPs have been used as convenient and efficient vehicles for drug delivery[29,30]. The exact mechanism of their intracellular cargo delivery is not yet fully understood, but the delivery was found to be mediated by endocytosis[27,31,32] and direct membrane penetration[27,33,34]. CPP delivery efficiency depends on physicochemical peptide properties[35,36], peptide modifications[37], cargo types[35,37], cell types[35], and the peptide/cargo ratio[38].

Several studies have been conducted to predict the penetration performance of CPPs[36,39]. The prediction models have been based on simple chemico–physical properties of the CPP sequences and designed for binary classification, which means that they predict whether or not, the peptide in query can penetrate cellular membranes in mammalian cells. Recently, a database of CPPs (CPPsite 2.0) was constructed that provides information about the experimental evaluation results in mammalian cells of 1855 CPPs with various types of cargo[37]. CPPs, therefore, have been widely employed in mammalian biotechnology as they are highly efficient in membrane penetration. For example, CPP-conjugated transcription activator-like effector nucleases (TALENs) and CPP-mediated Cas9/sgRNA delivery systems have been reported to improve the efficiency of genome modification with reduced off-target effects in various human cell lines[6,40].

In addition to their utility in mammalian biotechnology, CPPs can potentially be used in bacterial engineering to alter phenotypes by directly delivering functional proteins and to regulate the expression of biologically functional proteins without plasmids. However, over the past few decades, CPP research in bacteria has been focused on the anti-bacterial properties of CPPs in drug-resistant pathogenic bacteria[41–44]. Only a few studies have developed that CPPs are able to deliver protein cargo into bacteria[45], and CPP efficiency in bacteria is much lower than that in mammalian cells. However, with the increasing demand for microbial metabolic engineering and biotechnology, CPPs have attracted great interest as a potential plasmid-free engineering tool for delivering functional proteins into bacteria. For example, exogenous CPP/GFP complexes have been shown to be internalized without cytotoxic effects in cyanobacteria[45]. In addition, a small CPP library was screened in *Escherichia coli* to investigate cell penetration efficiency and cytotoxicity[46].

In this study, we aimed to identify efficient CPPs in *E. coli* and develop a method for improved delivery efficiency. Subsequently, we used the CPPs to remove plasmids from *E. coli* cells by delivering I-SceI restriction enzymes. We also developed a plasmid-free method of excising a marker gene integrated into the genome

of *Methylomonas* sp. DH-1 by delivering Cre recombinase. *Methylomonas* sp. DH-1 has recently been shown to utilize methane as a sole carbon source and is gaining great interest as a platform for metabolic engineering[47–49]. However, the metabolic engineering of this methanotroph is challenging due to the lack of genetic manipulation tools, including the absence of artificial plasmids for genetic engineering. The schematic illustration of our study is depicted in Fig. 1. The CPPs identified in this study may facilitate the advance of bacterial biotechnology such as synthetic biology and metabolic engineering[50] by enabling plasmid-free engineering without a non-specific or unintended genetic modification.

## Results

**Improved delivery efficiency of CPP conjugates by electroporation in *E. coli*.** In bacteria, CPPs have been used without additional treatments, but recently, the chemical treatment of cells has improved the delivery efficiency of CPP conjugates[46]. Electroporation has been widely used as an effective method for biomacromolecule delivery, and electroporation is generally more efficient than chemical treatments, such as that with $CaCl_2$[51]. In this study, we developed and optimized a new electroporation-based delivery method to further improve CPP delivery efficiency, and used the method for CPP delivery experiments.

For the evaluation and optimization of the electroporation-based method, five efficient CPPs were selected from the previous reports[45,46] (Fig. 2a). The CPP were labeled with TAMRA to trace their penetration into the cytoplasm. First, we prepared electrocompetent and chemically competent cells[46]. As a negative control, we also prepared non-treated cells. After treating the five TAMRA-labeled CPP peptides, either distilled water or 20 mM Tris-Cl (pH = 7.5) was added as a buffer to the *E. coli*–CPP mixture and incubated at 37 °C for 1 h. After washing the cells, their fluorescence intensities were measured by flow cytometry to quantify CPP delivery efficiency.

As shown in Fig. 2b, the electroporation-based method resulted in higher efficiency than the chemical treatment method, in which $CaCl_2$ was used to permeabilize the membrane. Interestingly, not only the method but also the choice of buffer affected CPP penetration efficiency. For the chemical treatment method, distilled water was associated with increased delivery efficiency. For the electroporation-based method, the use of Tris-Cl buffer (pH = 7.5) was associated with increased delivery efficiency. Consequently, the use of electroporation and Tris-Cl buffer (pH = 7.5) resulted in a delivery of CPP conjugates into *E. coli* that improved by 78.7-fold and 23.9-fold relative to that of the control and chemical treatment method, respectively. Therefore, the electroporation-based method with Tris-Cl buffer was used throughout this study.

**CPP library construction.** We collected CPPs from the CPPsite 2.0[37], which includes eukaryotic CPPs, from literature[46], and from several manually modified variants. The five CPPs (No. 14, 15, 16, 17, and 24) used for the evaluation of electroporation-based CPP penetration efficiency were also included in the library. Our constructed library contains 98 CPPs, and to our knowledge, it is the largest screening library of CPPs evaluated in *E. coli*. The whole list of CPPs and their physicochemical properties are shown in Supplementary Data 2.

**CPP penetration efficiency evaluation.** The cell penetration efficiencies of the CPPs were measured by the attached TAMRA. As shown in Fig. 3a, the intensities of TAMRA-labeled CPPs that penetrated *E. coli* cells ranged from 13 to 4043 A.U. (~1–300 fold) when measured by flow cytometry (Fig. 3a and Supplementary

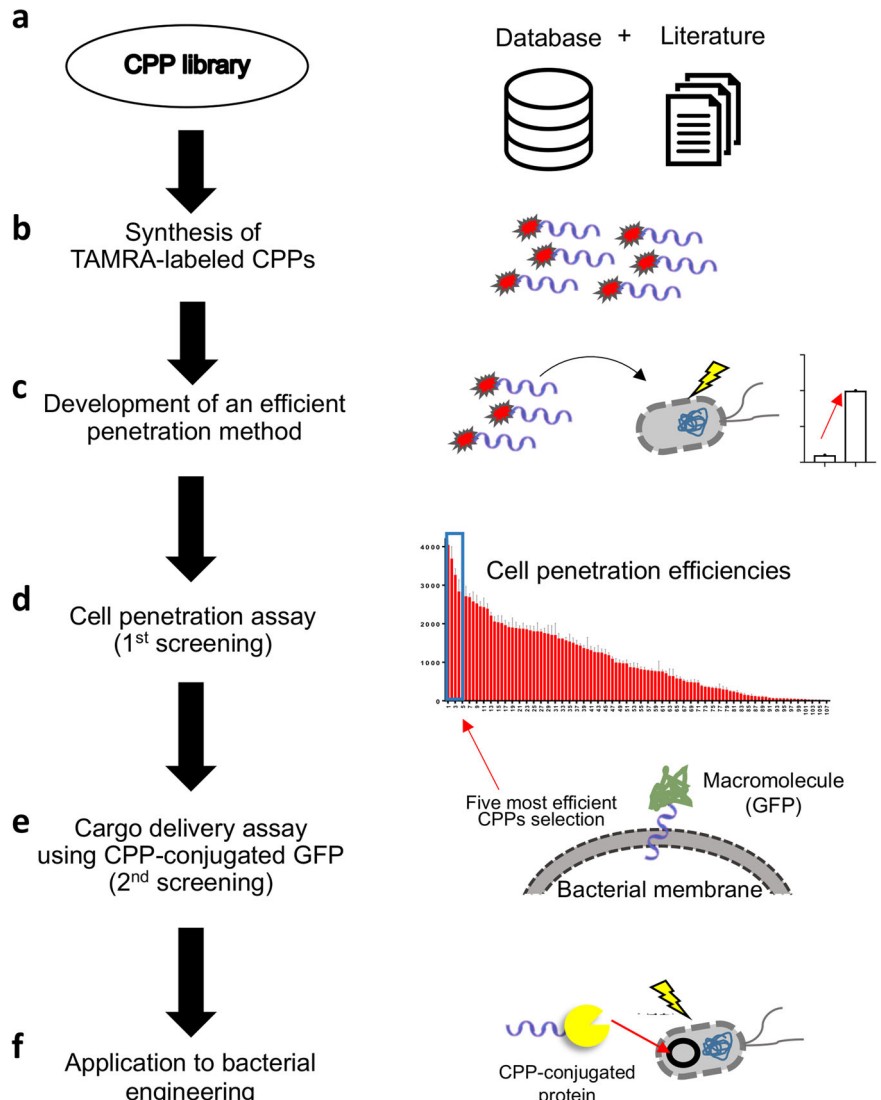

**Fig. 1 The schematic illustration of our strategy for finding efficient prokaryotic CPPs with applications in microbial biotechnology. a** We constructed a library of 98 CPPs from the literature and online databases. **b** For screening and the development of an efficient CPP delivery method, TAMRA-labeled CPPs were synthesized. **c** A new method for the improved delivery of CPP-conjugated proteins was developed that was suitable for bacterial engineering applications. **d** The library of CPPs was screened to identify the most efficient CPPs in *E. coli*. **e** The five most efficient CPPs in *E. coli* in terms of penetration efficiency and cytotoxicity were selected, and their efficiencies for delivering GFP cargo were measured. **f** The final two selected CPPs were used for bacterial engineering applications including plasmid removal from live *E. coli* cells using CPP-conjugated I-SceI and marker gene excision in *Methylomonas* sp. DH-1 using CPP-conjugated Cre recombinase.

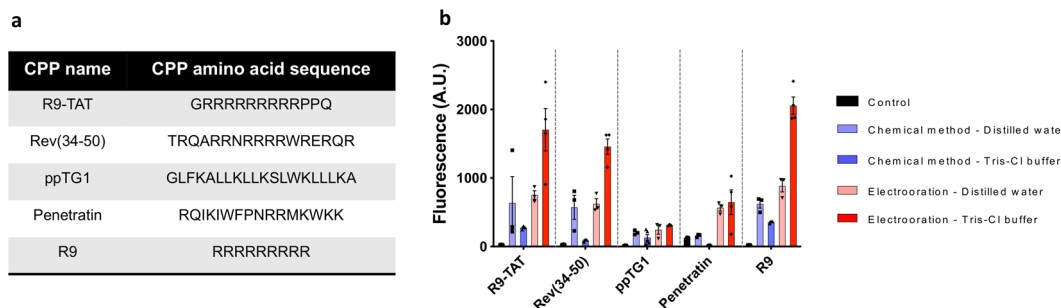

**Fig. 2 Electroporation-based method improved the penetration efficiencies of TAMRA-labeled CPPs. a** The list of the five selected CPP sequences used to evaluate the efficiencies of different treatment methods and incubation buffers. **b** The cell penetration efficiencies of the five CPPs measured by flow cytometry (in arbitrary units; A.U.). The CPPs were treated with either chemicals or electroporation with two different buffers (distilled water or Tris-Cl buffer at pH = 7.5). The mean and standard error were calculated from three independent experiments (*n* = 3–5). The dataset used to draw the graph is compiled in Supplementary Data 1.

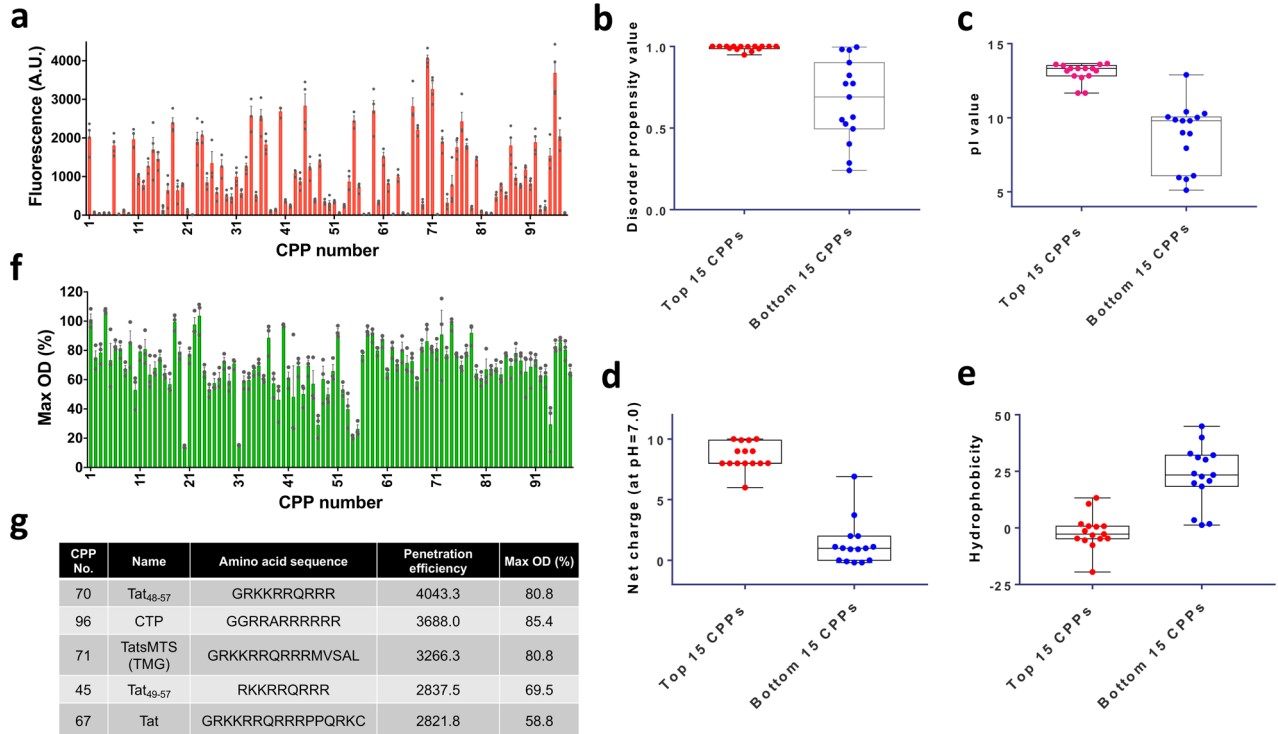

**Fig. 3 Penetration efficiencies and cytotoxic effects of the CPPs in the library and the distinct features of highly efficient CPP sequences. a** Since the CPPs were conjugated with TAMRA, the TAMRA fluorescence intensities within the cells indicate the cell membrane penetration efficiencies of the CPPs ($n = 3–5$). **b–e** Property comparison between the top 15 CPPs (efficient CPPs) and bottom 15 CPPs (inefficient CPPs). In the $t$ test, the $p$ values of the four physicochemical properties were lower than 0.0001. **b** Average disorder propensity value[†]. **c** Average pI value[‡]. **d** Average net charge at pH = 7.0[¶]. **e** Average hydrophobicity value of CPP sequences[§]. **f** The effect of the CPPs on *E. coli* cell growth (indicating cytotoxicity). *E. coli* cells treated with CPPs were grown to the stationary phase. Max OD$_{600}$ denotes the maximum optical density (OD) of CPP-treated cells divided by the maximum OD of control cells. The mean and standard error were calculated from three independent experiments ($n = 3$). **g** The list of selected efficient and safe CPPs. The datasets used to draw the graphs are compiled in Supplementary Data 1. Dagger symbol, https://iupred2a.elte.hu/plot Double dagger symbol, https://www.biosyn.com/peptidepropertycalculator/peptidepropertycalculator.aspx Paragraph symbol, https://www.biosyn.com/peptidepropertycalculator/peptidepropertycalculator.aspx Section sign, https://www.thermofisher.com/kr/ko/home/life-science/protein-biology/peptides-proteins/custom-peptide-synthesis-services/peptide-analyzing-tool.html.

Data 2). Interestingly, six out of the top ten high-efficiency CPPs were derived from the HIV Tat sequence (Supplementary Data 2), which are already known to have the capability to penetrate mammalian cells[37].

To investigate the distinct features of the highly efficient CPPs, the peptide sequences of the top 15 and bottom 15 CPPs were analyzed. The disorder propensity values of the efficient CPPs were all close to 1, which means that the peptides were structurally flexible rather than rigid. Conversely, the disorder propensity values of the inefficient CPPs varied (Fig. 3b). This indicates that the flexibility helped the CPPs penetrate cell membranes. In addition, the pI values and net charges (at pH = 7.0) of the efficient CPPs were higher than those of the inefficient CPPs (Fig. 3c, d), which is consistent with the previous reports that cationic CPPs are more efficient than neutral and anionic CPPs and that positively charged CPPs readily bind to the negatively charged phospholipids and lipopolysaccharides of bacterial cell membranes[46]. We also found that the efficient CPPs were less hydrophobic, while the inefficient CPPs were more hydrophobic (Fig. 3e). This may be because electroporation can transfer hydrophilic and charged molecules to cells by forming aqueous hydrophilic pores in the membranes[52–54].

**Evaluation of the anti-bacterial activity of CPPs**. CPPs often possess anti-bacterial properties[41–43]. After investigating the cell penetration capacity, to identify non-cytotoxic CPPs for bacterial

engineering, the potential anti-bacterial activities of the CPPs were evaluated by measuring *E. coli* growth after treatment with CPPs. The changes in growth rate of the cells treated with CPPs are shown in Fig. 3f, and the growth curves of cells treated with CPPs are shown in Supplementary Fig. 1. The changes in the growth (maximal optical density) ranged from 14.01% to 107.01% relative to the growth of the control (non-CPP-treated cells). The anti-bacterial activity was categorized into two classes (cytotoxic and safe) with a cutoff of 50% (Supplementary Data 2). Of the 98 CPPs, 10 exhibited significant cytotoxicity and 88 exhibited less or insignificant cytotoxicity. From these screening results, we were able to identify five CPPs that were highly efficient and safe (Fig. 3g).

We also investigated four features (disorder propensity, pI, net charge, and hydrophobicity) of the top 15 and bottom 15 cytotoxic CPPs. However, there were no significant differences in these features (Supplementary Fig. 2a–d). Furthermore, we could not find any statistical correlation between cell permeability and cytotoxicity (Supplementary Fig. 2e).

**Confirmation of CPP penetration of bacterial cytoplasm**. The efficient CPPs had many positive charges to mediate their binding to bacterial cell membranes. However, it was unclear if the CPPs had penetrated the membrane and entered the cytoplasm or if they were merely attached to it. To confirm their penetration, we selected the top five efficient CPPs, which were also less cytotoxic

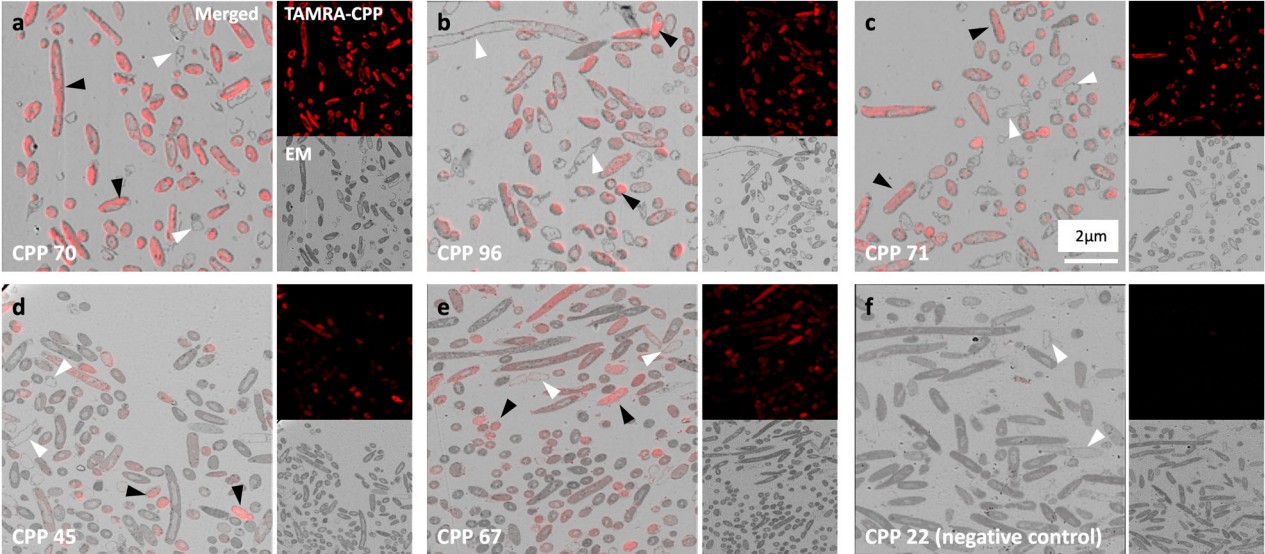

**Fig. 4 Electromicroscopic pictures of *E. coli* cells treated with the top five highly efficient CPPs.** Each image set shows a fluorescence image (TAMRA-labeled CPP, top right), an electron microscope image (EM, bottom right), and a merged CLEM image. **a–e** CPPs that were efficient in penetration. **f** CPP 22 was inefficient in penetration and was selected as a negative control. Black arrowheads indicate the bacteria in which the cytoplasm was intact and TAMRA-labeled CPPs emitted a red color. White arrowheads indicate bacteria in which the cytoplasm was removed during sectioning and that therefore lacked CPPs.

than the inefficient CPPs. Correlative light and electron microscopy (CLEM) was used to confirm that the CPPs crossed the membrane and entered the bacterial cytoplasm. CLEM is a combinational technique of light microscopy (LM) and electron microscopy (EM). We observed the same bacterial region under both a fluorescence microscope (FM) and scanning electron microscope (SEM).

Consequently, it was confirmed that TAMRA-labeled CPPs were present within the microstructures of bacteria. As shown in Fig. 4, the top five TAMRA-labeled CPPs successfully penetrated the *E. coli* cells. Preparing ultrathin sections (a thickness of 50 nm) enabled us to distinguish the cytoplasm from the cell membrane. The sectioned microscope images allowed us to identify which regions contained bacterial cytoplasm and which did not. The location of CPPs was identified by analyzing the structure of bacteria in the CLEM images. If the CPPs were only bound to the cell membrane, the membrane region would emit a red color indicating the bound TAMRA. However, most of the red color was observed in cells in which the bacterial cytoplasm was intact (black arrowheads in the CLEM images).

**Computational modeling to predict CPP penetration efficiency.** Using the penetration efficiencies of 98 CPPs, we developed a penetration efficiency prediction model based on the random forest algorithm. First, the peptide features (amino acid compositions, dipeptide compositions, and AAIndex values) were calculated[55]. The feature values were converted to log scale and normalized. For feature selection, each Pearson's correlation coefficient between feature and penetration efficiency was calculated. Interestingly, many features that were related to net charge ($|r| = 0.632$) and hydrophobicity ($|r| = 0.627$) were top ranked, which is consistent with our CPP analysis results and indicates that net charge and hydrophobicity are important factors of CPP penetration efficiency.

We built models using the top *n* features (50, 100, 150, 200, 250, 300, 350, 400, 450, and 500), and the model trained with the top 250 features produced the best performance (Supplementary Fig. 3a). Thus, we developed a penetration efficiency prediction model trained with the top 250 features. When 10-fold cross-

validation was performed, the model produced values of $R^2 = 0.591$ and root mean square error (RMSE) = 0.405. The 10-fold cross-validation result is shown in a scatter plot (Supplementary Fig. 3b). When the model was changed to binary classification (efficient or inefficient categories) with a cutoff value of 2.5 (log-scaled penetration efficiency), the model exhibited an 82.1% accuracy.

To design new efficient CPPs, we generated random peptide sequences with 10–20 amino acids and generated combinatorial sequences of the CPP sequences in the training dataset. As a result, nine CPPs were predicted to have the highest penetration efficiencies, and therefore the penetration efficiency and cytotoxicity of the nine CPPs were evaluated (Fig. 5) by fusing the CPPs with TAMRA. The nine predicted CPPs exhibited sufficiently high penetration efficiencies to be categorized as efficient CPPs. Although their efficiencies were mostly high, they were not higher than that of the best CPP in the library (Figs. 3a, 5b). Nonetheless, the model may still be useful for finding and designing efficient CPPs. We also measured the cytotoxicity of the nine CPPs. The growth rate changes of *E. coli* cells treated with the nine CPPs ranged from 64.20% to 125.83% (Fig. 5c), indicating relatively low cytotoxicity.

To build a more accurate model, extensive CPP evaluation data are required. Thus, there is a demand for building a public database of bacterial CPPs to advance the fields of CPP-based model development and CPP-based bacterial engineering.

**Protein delivery efficiency of the top five CPPs.** In mammalian cells, the cell penetration efficiency of CPPs depends on cargo type[35,37]. Thus, we also confirmed that the five selected efficient CPPs were consistently efficient when conjugated with proteins. To construct CPP-conjugated GFP expression plasmids, each CPP was codon optimized for better expression in *E. coli* and each CPP with a Gly–Ala linker was inserted into the N-terminus of the *gfp* gene (Fig. 6a). All plasmids were individually transformed into *E. coli* BL21(DE3) cells, and CPP-conjugated GFP proteins were purified.

When *E. coli* cells were treated for 1 h either with each purified CPP-conjugated GFP protein or with just GFP protein as a

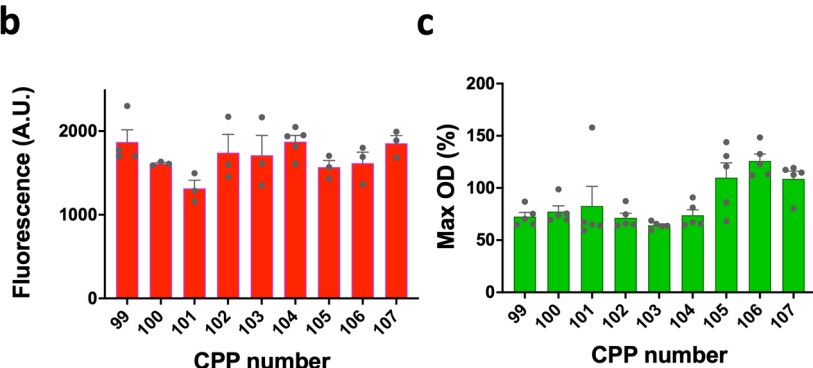

**a**

| CPP No. | Amino acid sequence | Penetration efficiency | Max OD (%) |
|---|---|---|---|
| 99 | CVSRRRRRRGGRRRRGRKKRRQRRRNSNFN | 1869.9 | 72.6 |
| 100 | RRRRRRRRRGRKKRRQRRRPPQRKC | 1611.5 | 77.4 |
| 101 | RRRRRRRRRRRRRRRRR | 1314.9 | 82.7 |
| 102 | RKKRRQRRRRRRRRRRRR | 1743.2 | 71.4 |
| 103 | GRKKRRQRRRPPQRKCGGRRARRRRRR | 1710.5 | 64.2 |
| 104 | VSRRRRRRGGRRRRRPQNFPNNNA | 1873.5 | 74.0 |
| 105 | VSRRRRRRGGRRRRHNNHR | 1569.7 | 110.0 |
| 106 | GGRRARRRRRRSPPPP | 1617.7 | 125.8 |
| 107 | RRRRRRRRRPPPSH | 1855.8 | 108.8 |

**Fig. 5 The nine designed CPP sequences and their penetration efficiency and cytotoxicity. a** The list of nine CPP sequences that were predicted by our developed computational model to have highly efficient cell membrane penetration. **b, c** Evaluated penetration efficiency and cytotoxicity of the TAMRA-labeled CPPs, respectively. Biological replicates ($n$): $n = 3–5$ for penetration efficiency, $n = 5$ for cytotoxicity. Max OD (%) denotes the maximal OD of CPP-treated cells divided by the maximal OD of non-treated cells, multiplied by 100. The datasets used to draw the graphs are compiled in Supplementary Data 1.

control, the fluorescence intensities of the cells treated with CPP 70 (Tat$_{48–57}$)-conjugated GFP and CPP 45 (Tat$_{49–57}$)-conjugated GFP increased by 12-fold and 10-fold, respectively, compared with that of the control (Fig. 6b). However, the other three CPPs did not exhibit a significant increase in GFP delivery. This result indicates that a small difference in the amino acid sequence of the CPP can greatly affect protein delivery efficiency (Fig. 3g). We also investigated the association of CPP incubation time with delivery efficiency. When cells were incubated with CPP-conjugated GFP for 1–4 h after electroporation, the protein delivery efficiency was saturated within 3 h of incubation (Supplementary Fig. 4). Consequently, we selected two efficient CPPs, CPP 70 and CPP 45, and used a 3-h treatment time for the bacterial engineering applications.

**Application to plasmid removal in E. coli.** In bacterial engineering, it is often necessary to remove unused plasmids after genome engineering, to study the physiology of plasmid-cured cells, or to reduce plasmid-mediated multidrug resistance of pathogenic bacteria, among other purposes. Traditional plasmid removal methods require treatment with mutagenic chemical agents such as ethidium bromide, sodium dodecyl sulfate, and acridine orange, which introduce mutations within plasmid nucleotides and disrupt plasmid replication[56]. This method may cause cell damage or introduce unwanted mutations to the genome. Another method uses replicon incompatibility[56] by introducing another plasmid that

has an incompatible replication origin with the origin of the pre-existing plasmid. The two plasmids compete with each other for replication, and the pre-existing plasmid may disappear. However, this method requires the subsequent removal of the introduced plasmid.

In this study, we developed a simple and easy-to-use plasmid removal method that does not require the addition of any plasmids or chemical agents. In this method, a meganuclease I-SceI is directly delivered into bacterial cells using the highly efficient CPPs, and the delivered I-SceI eliminates the target plasmid, which contains an I-SceI recognition site (Fig. 7a). Because the recognition sequence of I-SceI is 18-nt long, it is rarely found within the prokaryotic genome, including that of *E. coli*. The random occurrence probability of the restriction site is ~1/70 billion bp. To this end, we fused the two selected CPPs with the meganuclease I-SceI and inserted an I-SceI recognition sequence into a target plasmid.

To determine whether CPP fusion altered the activity of the I-SceI protein, we purified the CPP-conjugated I-SceI proteins from *E. coli* BL21(DE3) and assayed the in vitro cleavage activity of the two CPP-conjugated I-SceI proteins. As shown in Supplementary Fig. 5a, fusion with the CPP sequences inhibited the activity of the I-SceI protein. To reduce the structural effect of CPP sequences on the folding of I-SceI protein, we co-introduced the pGro7 plasmid expressing two chaperones, *gro*ES and *gro*EL, and the chaperones made the two CPP-conjugated I-SceI proteins biologically active (Supplementary Fig. 5a, b).

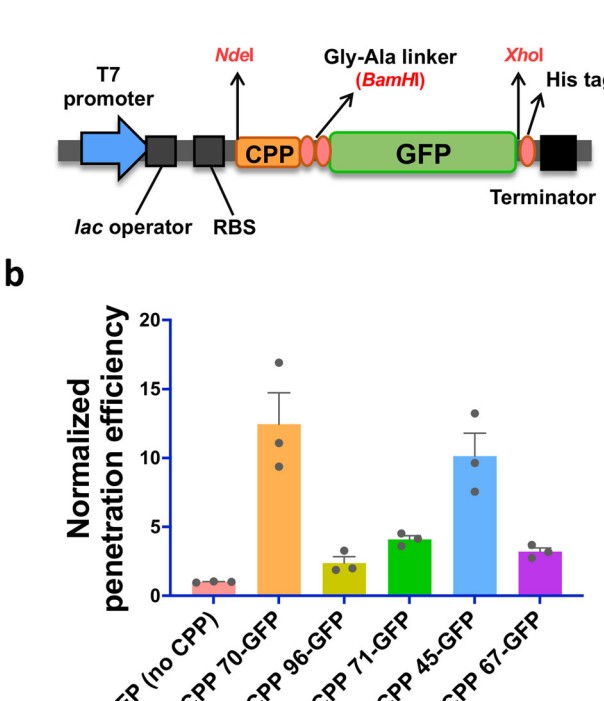

**Fig. 6 Penetration efficiencies of the five selected CPP-conjugated GFP proteins. a** The genetic structure designed to express CPP-conjugated GFP proteins in pET21a(+) plasmids, which contained T7 promoter-lac operator-RBS-CPP-(Gly–Ala linker)-GFP-(6 × His tag)-terminator. **b** Cell penetration assay results of the five purified CPP-conjugated GFP proteins. GFP intensities of cells were measured by flow cytometry after washing out extracellular CPP-conjugated GFP proteins. The protein cargo (GFP) delivery efficiencies of the CPPs were normalized by the efficiency of a control GFP protein (with no CPP). The mean and standard error were calculated from three independent experiments ($n = 3$). The dataset used to draw the graph is compiled in Supplementary Data 1.

Additionally, for the counterselection of plasmid-removed cells, we inserted the tetracycline resistance gene (tetA) into the plasmid to be removed. With the plasmid, the cells can survive in tetracycline media but not in $Ni^{2+}$ media. Conversely, without the plasmid, the cells cannot survive in tetracycline media but can survive in $Ni^{2+}$ media[57]. The tetA gene induces host cells to accumulate twice as much $Ni^{2+}$ as normal cells, so tetA-containing cells are more susceptible to $NiCl_2$ toxicity than normal cells[57]. To determine a suitable $Ni^{2+}$ concentration for the selection of plasmid-removed cells, wild-type E. coli cells and tetracycline-resistant cells harboring the tetA gene in the plasmid were grown on Luria–Bertani (LB) agar plates containing 1–4 mM $NiCl_2$. As shown in Supplementary Table 1, after incubation at 37 °C for 24 h, wild-type E. coli cells were capable of growing on LB agar plates containing up to 2.2 mM $NiCl_2$. In contrast, tetA-harboring cells were not able to grow on plates containing 2.0 mM $NiCl_2$. Therefore, we chose 2.0 mM $NiCl_2$ as the selection concentration (Supplementary Table 1).

Finally, the plasmid removal experiment was conducted with the two CPP-conjugated I-SceI proteins on the cells that carried the plasmid containing the I-SceI recognition sequence and tetA gene. The resulting efficiencies with respect to various replication origins (copy numbers) are shown in Fig. 7b. When the experiments were performed using cells carrying the plasmids

with a pSC101 origin (copy number: ~5 copies) or a ColE1 origin (copy number: 15–20 copies), the plasmid removal efficiencies of CPP 70-conjugated I-SceI and CPP 45-conjugated I-SceI were significantly improved, compared with that of the control, (I-SceI without CPP) by 3.8-fold and 3.4-fold, respectively, for the plasmid with a pSC101 origin, and by 4.1-fold and 3.4-fold, respectively, for the plasmid with a ColE1 origin. Even in the plasmid with a RSF1030 origin (>100 copies), there was a slight improvement in plasmid removal efficiency when CPP 70- and CPP 45-conjugated I-SceIs were used (1.6-fold and 1.3-fold, respectively). Our data demonstrated the dual function of CPP-conjugated I-SceI in cell penetration ability and DNA cleavage activity and its utility for bacterial plasmid curing.

**Application to marker gene excision in *Methylomonas* sp. DH-1.** To investigate whether the selected CPPs were able to deliver protein cargo to other bacterial species, we applied the CPPs to a methanotroph, *Methylomonas* sp. DH-1. *Methylomonas* sp. DH-1 can convert methane into various value-added compounds and thus has become a promising platform strain for metabolic engineering[47,49,58]. In genome integration, homologous recombination is commonly used in *Methylomonas* sp. DH-1, and an antibiotics marker gene is used for the selection of knocked-in cells. However, since currently there are no artificial plasmids for the genetic engineering of *Methylomonas* sp. DH-1, it is difficult to remove the marker gene. Thus, a protein delivery method is required for marker excision.

In this study, we developed a marker gene (antibiotic resistance gene) excision method by directly delivering a Cre recombinase that excises the marker gene flanked by loxP sites. Two CPP-conjugated Cre recombinases were constructed as fusion proteins. The in vitro activity assays are shown in Supplementary Fig. 6. For the counterselection of marker gene-excised cells, the tetA gene was also used. However, since *Methylomonas* sp. DH-1 exhibits resistance at low concentrations of tetracycline, the $kan^R$ gene was additionally used for efficient selection. Therefore, we constructed a genetic structure for homologous integration into the genome of *Methylomonas* sp. DH-1 that contained the $kan^R$ and tetA genes flanked by two loxP sites for the selection of marker gene-integrated cells and counterselection of marker gene-excised cells, respectively (Fig. 7c, d). In this study, as a proof-of-concept, only the antibiotic resistance genes were integrated into the genome by homologous recombination. For practical applications, other genes can also be integrated in the genome with the antibiotic resistance genes.

First, the antibiotic resistance genes flanked by loxP sites were integrated into the genome of *Methylomonas* sp. DH-1 by homologous recombination. In this step, kanamycin was used for selecting genome-engineered cells. Then, the engineered cells were treated with CPP-conjugated Cre recombinases to excise the antibiotic resistance genes ($kan^R$ and tetA). In this step, the marker-excised cells were selected on nitrate mineral salts (NMS) plates containing 4 mM $NiCl_2$, the optimal selection $Ni^{2+}$ concentration for *Methylomonas* sp. DH-1 (Supplementary Table 2). As shown in Fig. 7e, only the Cre recombinase conjugated with CPP 70 successfully excised the marker genes with high efficiency. Interestingly, the use of NMS media (the growth media of *Methylomonas* sp. DH-1), rather than Tris-Cl buffer, significantly improved marker excision efficiency. Taken together, these results indicate that the protein delivery efficiency of CPPs may vary depending on the bacterial species and buffers used. Nevertheless, our data demonstrate the broad applicability of CPPs to the delivery of functional proteins for genetic engineering in various bacterial species.

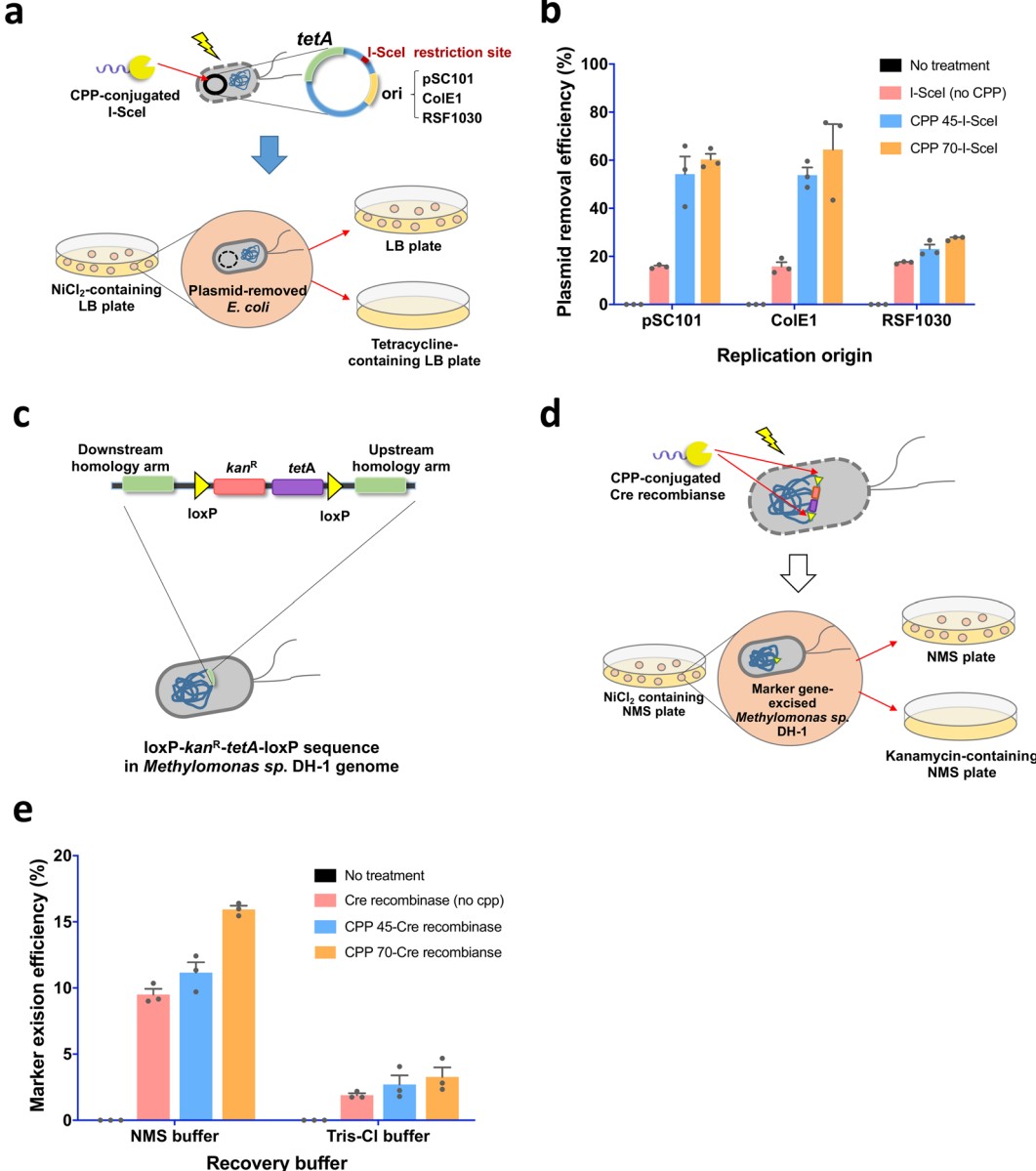

**Fig. 7 Plasmid removal using CPP-conjugated I-SceI in *E. coli* and marker gene excision from the genome of *Methylomonas* sp. DH-1 using CPP-conjugated Cre recombinase. a, b** Plasmid removal in *E. coli*. **c–e** Marker gene excision from the genome of *Methylomonas* sp. DH-1. **a** The strategy for CPP-based plasmid removal. The target plasmid to be removed contained an I-SceI recognition site, the tetracycline resistance gene for selection and counterselection, and three different replication origins (pSC101 origin, ~5 copies; ColE1 origin, 15–20 copies; RSF1030 origin, >100 copies). **b** Measured plasmid removal efficiencies using CPP-conjugated I-SceI proteins (*n* = 3). CPP 45 and CPP 70 had significantly increased efficiencies compared with that of the control, but there were no statistically significant differences between CPP 45 and CPP 70. **c** The DNA structure used for genomic integration into the genome of *Methylomonas* sp. DH-1. **d** The strategy for CPP-based marker gene excision from the genome of *Methylomonas* sp. DH-1. **e** Marker gene excision efficiencies of CPP-conjugated Cre recombinases (*n* = 3). Using Tris-Cl buffer did not result in significant efficiency differences among the control, CPP 45 and CPP 70. However, nitrate mineral salts (NMS) growth media resulted in a statistically different efficiency only for CPP 70. The datasets used to draw the graphs are compiled in Supplementary Data 1.

**CPP penetration efficiency and toxicity in mammalian cells (HEK293 and CHO cells).** To date, mostly CPPs have been studied and utilized in mammalian cells[37], but in this study, we studied CPPs in bacteria, *E. coli*. Thus, it would be very useful to find the similarities and differences of CPPs between prokaryotes and eukaryotes. For example, the CPPs derived from the HIV Tat sequence work well in both *E. coli* and mammalian cells[37,59], while CPP 63 (KLPVM) was known as a highly efficient CPP in mammalian cells[37,60], but showed low efficiency in *E. coli* (Fig. 3a).

For comparison, we selected ten highly efficient CPPs and ten highly inefficient CPPs from our study and evaluated their efficiencies in two different mammalian cells, HEK293 and Chinese Hamster Ovary (CHO) cells. As shown in Fig. 8a, generally the ten efficient CPPs in *E. coli* also showed high efficiency in HEK293 cells. Conversely, the efficient CPPs in *E. coli* showed low penetration efficiencies in CHO cells (Fig. 8b). Surprisingly, when their correlations were analyzed, we could not find any statistical correlations of the penetration efficiency among the three cells (Fig. 8c–e).

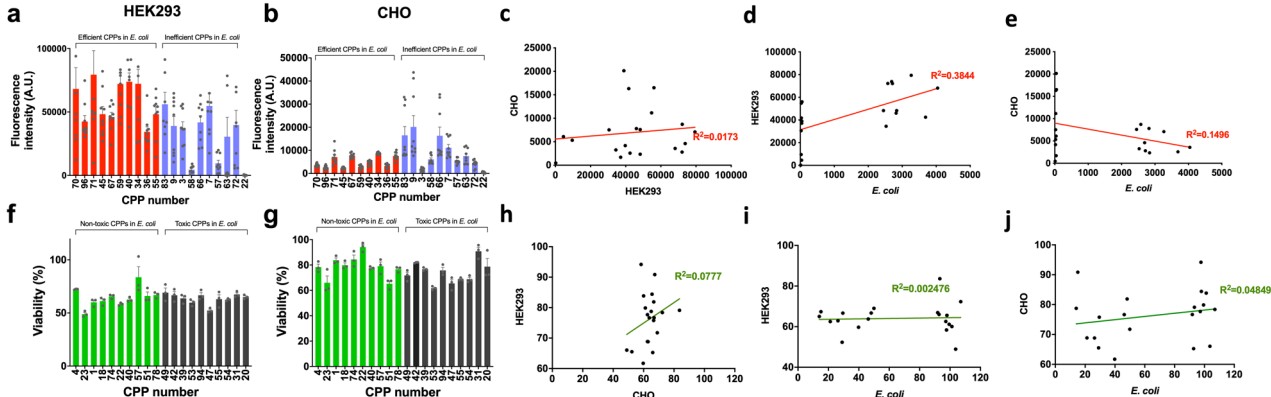

**Fig. 8 Penetration efficiencies and cytotoxic effects of the CPPs in HEK293 and CHO cells, and comparison of CPP effects in different cell types. a**, **b** Ten highly efficient CPPs and ten highly inefficient CPPs in *E. coli* were selected, and their cell penetration efficiency was measured in HEK293 and CHO cells ($n = 9$). **c–e** Linear correlations of the penetration efficiencies among the three cell types. **f**, **g** Ten highly toxic CPPs and ten highly safe CPPs in *E. coli* were selected, and their cytotoxicity was measured in HEK293 and CHO cells ($n = 3$). **h–j** Linear correlations of the cytotoxicity among the three cell types. The datasets used to draw the graphs are compiled in Supplementary Data 1.

For toxicity comparison, we selected ten highly toxic CPPs and ten highly safe CPPs, and measured their toxicities in HEK293 and CHO cells. Most of the CPPs did not show significant toxicity in both HEK293 and CHO cells, but CHO cells were more resistant against CPP toxicity than HEK293 cells (Fig. 8f, g). When correlations were analyzed, we also could not find any correlations among the three cells (Fig. 8h–j).

The low correlations represent that CPP penetration efficiency may be different from cells to cells, and this may hinder the development of universal CPPs that are efficient in many different cell types. However, the three cells used in this study are very different: prokaryotic cell (*E. coli*), human embryonic kidney cell (HEK293), and hamster ovary cell (CHO). Thus, more studies are required in order to find the common and distinct features of CPPs.

## Discussion

In this study, we constructed a library of CPPs and evaluated their cytotoxicity and efficiency in penetrating bacteria (*E. coli*), to deliver biologically functional proteins. Then, we developed a website containing the experimental data of the CPPs evaluated in bacteria, which is accessible at http://ssbio.cau.ac.kr/cpp (Supplementary Data 2). We also developed a method to further improve the penetration of CPPs through the bacterial membrane.

When the correlation between CPP residues and their penetration efficiency was analyzed, it was found that disorder propensity, net charge, pI, and hydrophobicity were highly correlated (Fig. 3b–e). These results were reaffirmed in the feature selection for modeling; features related to hydrophobicity and net charge had a high impact on the prediction model. Interestingly, of the top ten efficient CPPs, six were derived from the protein transduction domain of HIV-1 Tat (Supplementary Data 2). Thus, the sequences of the CPPs exhibited a high level of similarity. However, their cytotoxic effects were significantly altered by minor amino acid differences (Supplementary Data 2, see CPP 70 and CPP 55). Our analysis indicated that there may be no correlation between cell penetration efficiency and anti-bacterial effect (Supplementary Fig. 2e); rather, there may be other, different factors that affect anti-bacterial properties.

We also developed a computational model to predict CPP performance so as to design new CPPs. The model exhibited a high level of performance ($R^2 = 0.591$) (Supplementary Fig. 3b). When the model was adjusted to solve a binary classification

problem (to determine whether each CPP was efficient or inefficient), the model performed with high accuracy (82.1%). Using this model, nine new CPPs were designed, and their efficiency was evaluated. The newly designed CPPs exhibited high efficiency and low cytotoxicity (Fig. 5) but were not more suitable than the most efficient CPP in the library (Fig. 3a). The decreased efficiency might stem from the relatively longer CPP sequence in length[61]. Although the designed CPPs did not have the best efficiency, they were still highly efficient relative to other CPPs in the library. Thus, the model produced valid results when used to identify efficient CPP sequences. In the future, in order to build a more accurate prediction model, more experimental data for CPPs evaluated in bacteria are required, indicating a demand for the accumulation of CPP data and regular updates of our website.

To test the practical applications of CPPs in bacterial engineering, we developed methods for plasmid removal using CPP-conjugated I-SceI in *E. coli* and for marker gene excision from the genome using CPP-conjugated Cre recombinase in *Methylomonas* sp. DH-1. The developed methods successfully functioned as designed, and CPP-conjugated proteins performed with more efficiency than the proteins without CPPs (Fig. 7). However, there could be a potential folding issue when a CPP is fused to a protein since CPPs are generally are highly charged. For example, the purified CPP-conjugated I-SceI showed no activity without chaperone expression, while purified I-SceI showed normal restriction activity. Therefore, it should be considered whether CPP interferes with the folding of the cargo protein or not.

CPPs offer many advantages over conventional DNA-based genetic engineering strategies. CPP-based delivery systems do not require the use of any plasmids or expression inducers–and do not introduce any potential mutations to the bacterial genome, for example, by recombination. Furthermore, the functional proteins delivered by CPPs are able to act transiently, during the desired time period, via natural degradation through proteolysis. Therefore, CPPs may have many potential applications in bacterial engineering. Some examples are: introducing functional proteins transiently in physiology studies; engineering genes or genomes without plasmids; and controlling pathogenic bacteria with their cytotoxicity or by delivering toxic biomolecules that specifically target pathogens. Consequently, CPP-based biotechnology has several attractive features, such as the simplicity of experiments and applicability to a broad range of protein cargo and bacterial species.

Despite of the wide applicability of CPPs in biotechnological engineering, CPPs may show different properties (penetration

efficiency and toxicity) from cell types to types: one efficient CPP developed in one cell type may not work well in other cell types. This may require iterative screening for efficient CPPs in every cell type. However, since we used only three very distinct cell types (prokaryote, human kidney cell, and hamster ovary cell), more studies are required to draw a concrete conclusion.

In summary, we envision that CPPs will be widely applied, not only for the transport of bio-active protein cargo but also for bacterial biotechnology purposes. The methods developed and the CPPs identified in this study may facilitate advances in bacterial biotechnology.

## Methods

**CPP library preparation and synthesis**. A library containing 98 CPPs was compiled from a database of eukaryotic CPPs (CPPsite 2.0)[37] and the literature[46]. The CPPs in the library are listed in Supplementary Data 2. CPPsite 2.0 contains 1699 CPP sequences and their physicochemical properties. We selected CPPs according to the following criteria: (1) a length of 5–24 amino acids, (2) chiral L-form excluding CPP 80 (rrrrrrrrcqcrrkn), and (3) linearity. Fifty-seven CPPs were selected. In a prior report, 55 CPPs were evaluated in E. coli[46], so we selected 39 efficient CPPs from the previous report. In addition, two CPPs were manually designed. A total of 98 CPPs were chemically synthesized with the addition of TAMRA via the Lys side chain amine at the C-terminus[46] to indicate the location of CPPs and enable measurement of their penetration efficiencies by fluorescence intensity. The excitation and emission wavelengths of TAMRA are 546 nm and 579 nm, respectively.

### Molecular cloning
*Construction of CPP-conjugated GFP, I-SceI, and Cre recombinase expression plasmids*. We constructed plasmids for the expression and purification of CPP-conjugated recombinant proteins in E. coli BL21 (DE3). For CPP-conjugated GFP production, the codon-optimized CPP sequences were inserted into the N-terminus of the gfp gene in the pWA-GFP plasmid[62]. Then, the CPP-fused GFP genes were digested with NdeI and XhoI and cloned into the pET21a(+) plasmid, which ultimately contained T7 promoter-CPP-(Gly–Ala linker)-GFP-(6 × His tag) (Fig. 5a).

For CPP-conjugated I-SceI production, the gfp gene was replaced with the I-SceI gene, and the CPP-conjugated I-SceI protein was purified by 6 × His tag. The I-SceI gene was PCR-cloned from the pBAD-I-SceI plasmid (Addgene #60960). As the purified CPP-conjugated I-SceI protein was not active (Supplementary Fig. 5a), the pGro7 plasmid (Takara #3340) was co-introduced into the E. coli harboring the CPP-conjugated I-SceI gene-containing plasmid. The pGro7 plasmid expresses two chaperones, GroES and GroEL. These chaperones assist the stable folding of the CPP-conjugated I-SceI protein, and their presence ensured that the purified CPP-conjugated I-SceI protein was biologically active (Supplementary Fig. 5b).

For the production of CPP-conjugated Cre recombinase (the Cre gene originated from Addgene #62730), the gfp gene was also replaced with the CPP-conjugated Cre recombinase gene. However, the expressed CPP-Cre protein formed inclusion bodies. To solubilize the CPP-Cre protein, the gene was transferred into the pTXB1 plasmid (New England BioLabs #N6707S) via the Ligation Independent Cloning (LIC) technique using T4 DNA polymerase (New England BioLabs, Massachusetts, USA). The pTXB1 plasmid contains a C-terminal chitin-binding domain (CBD) and intein tag.

*Construction of target plasmids to be removed by CPP-conjugated I-SceI*. To investigate the applicability of CPPs in plasmid removal using a CPP-conjugated I-SceI restriction enzyme, we constructed three different target plasmids containing a different replication origin: a pSC101 origin (low-copy number), ColE1 origin (mid-copy number), and RSF1030 origin (high-copy number). Then, an I-SceI recognition site was inserted into the target plasmids. In addition, for the selection of plasmid-containing cells and counterselection of plasmid-free cells, the tetA gene (tetracycline resistance gene) was introduced into the target plasmids. The selection and counterselection were conducted by using tetracycline and NiCl₂, respectively. The constructed plasmid structure is shown in Fig. 7a.

*Construction of a plasmid for genomic integration into Methylomonas sp. DH-1*. The pIns plasmid, which carries two 1 kb-long homology arms for genomic integration into Methylomonas sp. DH-1, was provided by Dr. Hahn (Institute of Chemical Processes, Seoul National University)[47]. We additionally inserted the kanᴿ and tetA genes flanked by two loxP sites for Cre-mediated excision. The kanᴿ gene was used for the selection of genome-integrated cells, and the tetA gene was used for the selection of marker gene-excised cells because Methylomonas sp. DH-1 is resistant to tetracycline[58]. The plasmid (pIns-LKTL) map is shown in Fig. 7c.

### Bacterial strains and growth conditions
**Bacterial strains and growth conditions**. E. coli DH5α was used for the DNA cloning and plasmid removal experiments. E. coli BL21 (DE3) was utilized as a host

strain for the expression of CPP-conjugated recombinant proteins. LB medium, containing 10 g of tryptone, 5 g of yeast extract, and 10 g of sodium chloride per liter, was used for the cultivation of E. coli strains, with appropriate selection antibiotics (100 µg/mL of ampicillin, 15 µg/mL of tetracycline, or 25 µg/mL of chloramphenicol).

Methylomonas sp. DH-1 (KCTC13004BP) was used for the marker gene excision experiment. Methylomonas sp. DH-1 was cultured in NMS medium[49] supplemented with 20% (v/v) methane at 30 °C with shaking at 170 rpm. For genetic integration experiments, the cells were transformed with a plasmid containing homology arms by electroporation[47].

For the expression of CPP-conjugated recombinant proteins, E. coli cells were grown overnight in a test tube at 37 °C, with shaking at 200 rpm. Next, 1/100 (v/v) of the cultured cells were transferred into fresh LB media supplemented with 100 µg/mL of ampicillin. When the cells reached an optical density at 600 nm (OD₆₀₀) of 0.5, the expression of CPP-conjugated proteins was induced by adding 0.4 mM of IPTG. After 5 h, the cells were harvested by centrifugation at 10,000 rpm for 10 min at 4 °C.

For CPP-conjugated I-SceI production, E. coli cells with the pGro7 plasmid and pET21a(+)-CPP-fused I-SceI were grown in fresh LB media containing 20 µg/mL of chloramphenicol and 0.5 mg/mL of L-arabinose. The arabinose was used to induce the expression of chaperones prior to IPTG induction of CPP-conjugated I-SceI expression.

### Purification of CPP-conjugated recombinant proteins
**Purification of CPP-conjugated recombinant proteins**. After IPTG induction, harvested cells were resuspended in lysis buffer A (50 mM Tris-HCl at pH 8.0, 100 mM NaCl, 5 mM imidazole, 1 mM phenylmethylsulfonyl fluoride) for His-tagged proteins (GFP and I-SceI) or in lysis buffer B (20 mM Tris-HCl at pH 8.5, 0.5 M NaCl, 1 mM EDTA) for CBD-intein-tagged proteins (Cre recombinase), and the cells were disrupted by sonication. The supernatant and pellet were collected separately and analyzed by sodium dodecyl sulfate-polyacrylamide gel electrophoresis (SDS-PAGE). CPP-conjugated recombinant proteins were purified from the supernatant, according to the manufacturer's instruction, by 6 × His tag (QIAGEN, Hilden, Germany) or CBD-intein tag (New England BioLabs, Massachusetts, USA).

Buffers of eluted CPP-conjugated proteins were exchanged by dialysis using a cellulose ester membrane with a molecular weight cutoff of 6–8 kDa (Spectrum Laboratories, Piscataway, NJ, USA). The CPP-conjugated GFP was stored in 1 × phosphate-buffered saline (PBS) buffer, and the CPP-conjugated I-SceI and Cre recombinase were stored in the manufacturer's recommended storage buffers (New England BioLabs, Massachusetts, USA).

### CPP penetration efficiency measurement
**CPP penetration efficiency measurement**. To compare CPP delivery methods (the chemical treatment approach and electroporation-based approach), wild-type DH5α cells were prepared as follows: Briefly, for the chemical treatment method[46], cells were treated with 50 mM CaCl₂ at an OD₆₀₀ of 0.7 and centrifuged at 5000 rpm for 10 min at 4 °C. Then, the cells were washed and centrifuged again using 50 mM CaCl₂ and resuspended in 4 mL of 50 mM CaCl₂ containing 15% glycerol. The cells were aliquoted, frozen using liquid N₂, and stored at −80 °C until use. Prepared cells at a density of 5 × 10⁷ cells/50 µL were treated with 5 µg of TAMRA-labeled CPPs (0.1 µg/µL) and incubated for 1 h at room temperature. After incubation, the cells were washed with 1 × PBS and centrifuged at 7000 rpm for 10 min three times to remove extracellular CPPs.

For the electroporation-based method, electrocompetent cells were prepared as the protocol reported in New England Biolabs (https://international.neb.com/protocols/2012/06/21/making-your-own-electrocompetent-cells). The prepared cells were diluted to a density of 5 × 10⁷ cells/50 µL using 10% glycerol, and 5 µg of TAMRA-labeled CPPs (0.1 µg/µL) was added to the cells. The E. coli–CPP mixture was transferred to a pre-chilled electroporation cuvette with a 0.2 cm gap, and electroporation was performed at 1.8 kV using an electroporator (MicroPulser, Bio-Rad). The cells were immediately recovered with buffer (either 20 mM Tris-HCl at pH 7.5, or distilled water) and incubated for 1 h at room temperature. Then, the cells were washed with 1 × PBS and centrifuged at 7000 rpm for 10 min three times. The fluorescence intensity of the CPPs in E. coli cells was quantitatively measured using a Guava EasyCyte flow cytometer (Millipore, Darmstadt, Germany) with an excitation wavelength of 547 nm and an emission wavelength of 580 nm.

For CPP-conjugated GFP delivery into E. coli, prepared cells at a density of 5 × 10⁷ cells/50 µL were mixed with 50 µL of purified CPP-conjugated GFP proteins (0.05 µg/µL) instead of TAMRA-labeled CPPs (0.1 µg/µL). In the case of CPP-conjugated protein treatment, the treatment concentration was reduced due to the limited capacity of the cuvette used for electroporation. After 1 h of incubation and three washes, the fluorescence intensities emitted from the cells were measured by using a flow cytometer (excitation at 488 nm; emission was collected with a 525/30 nm bandpass filter).

### Inhibition of bacterial cell growth by CPPs
**Inhibition of bacterial cell growth by CPPs**. E. coli DH5α cells were grown until OD₆₀₀ = 1 at 37 °C with shaking at 200 rpm. The cells were then diluted to 1/1000 with fresh LB medium and treated with 5 µg of TAMRA-labeled CPPs (0.1 µg/µL). Then, cell growth (OD₆₀₀) was monitored for 18 h by using the Synergy™ H1

hybrid multi-mode microplate reader (BioTek, Vermont, USA). As a control, we used *E. coli* cells that were not treated with CPPs.

**CLEM sample preparation**. TAMRA-labeled CPP-treated bacterial cells were fixed with 4% paraformaldehyde and 0.1% glutaraldehyde in 1 × PBS solution for 1 h. Fixed cells were washed with 0.1 M cacodylate buffer five times for 5–10 min and then incubated overnight in 1% uranyl acetate aqueous solution at 4 °C. The next day, incubated cells were washed with distilled water (five times for 5–10 min). After washing, samples were dehydrated through a graded ethanol series (30, 50, 70, 95, and 100%; each for 15 min at 4 °C) and 100% ethanol at room temperature. To infiltrate resins with bacteria cells, the following solutions were used sequentially: (1) ethanol: propylene oxide mixture solution (1:1; 30 min, twice), (2) propylene oxide, (3) propylene oxide:Epon 812 mixture solution (3:1, 1:1, 1:3; for 1 h each), and (4) pure Epon 812 (twice). Infiltrated samples were placed in embedding molds in an oven at 60 °C for 24 to 48 h.

**Bacterial cell imaging**. The bacterial EM blocks were sectioned by ultramicrotome (MT XL microtome, RMC Boeckeler, installed in the KIST Bio-imaging core facility) to a thickness of 50 nm and placed on a conductive plate for FM and SEM observation. Fluorescence images were taken by confocal microscope (LSM 800, Carl Zeiss) and deconvolution microscope (Thunder, Leica) using 63×. After fluorescence imaging, SEM images were taken in the same locations in which the fluorescence images of each sample had been taken. SEM (Teneo VS, FEI) images were taken at the following settings: 2KeV, 0.1 nA, T1 detector, and 5 nm pixel resolution.

**Computational modeling using the penetration efficiencies of 98 CPPs**. We constructed a random forest model to predict the penetration efficiency of a query peptide sequence. For the features, we used amino acid compositions, dipeptide compositions, and features calculated using AAIndex[55]. The total number of features was 505. The feature values were converted to log scale and normalized. For feature selection, we calculated the Pearson's correlation coefficients between the features and penetration efficiencies. When the top *n* features with a high Pearson's correlation coefficient (50, 100, 150, 200, 250, 300, 350, 400, 450, and 500) were trained, the model with 250 features produced the highest level of performance in a 10-fold cross-validation. Thus, the model was constructed by training with the 250 features.

To design new CPP sequences, we generated random peptides of 10–20 residues or combined the sequences in the library. As a result, we generated >200,000 peptides and selected nine with the highest prediction scores. We used the Orange software, version 3.22, to build the penetration efficiency prediction model[63].

**In vitro activity assays of CPP-conjugated I-SceI and Cre recombinase**. For the in vitro DNA cleavage assay of CPP-conjugated I-SceI, a *Not*I-linearized pGPS2 plasmid (New England BioLabs, Massachusetts, USA) containing an I-SceI recognition site was used as a substrate. Briefly, 500 ng of linearized plasmid DNA was mixed with 0.025–0.4 ng of purified CPP-conjugated I-SceI enzymes or 1 unit of commercially available I-SceI enzyme (New England BioLabs, Massachusetts, USA) as a positive control. The mixtures were reacted in 1 × CutSmart buffer provided by New England BioLabs at 37 °C for 1 h. The reactions were then halted by heating at 65 °C for 20 min. The cleaved products were visualized on ethidium bromide-stained agarose gels.

For the in vitro activity assay of CPP-conjugated Cre recombinase, we used the pIns-LKTL plasmid as a substrate, which was constructed for genomic integration into *Methylomonas* sp. DH-1 by homologous recombination and contained two loxP sites. One microgram of *Not*I-linearized plasmid DNA was mixed with 0.025–0.1 ng of purified Cre recombinase or 1 unit of commercial Cre recombinase (New England BioLabs, Massachusetts, USA) as a positive control. The mixtures were reacted in 1 × Cre reaction buffer provided by New England BioLabs at 37 °C for 1 h. The reaction was then halted by heating at 70 °C for 10 min. The cleaved products were visualized on ethidium bromide-stained agarose gels.

**Optimal Ni$^{2+}$ concentration determination for counterselection of *tetA*-removed cells**. To investigate the susceptibility of *tetA*-harboring *E. coli* and *tetA*-harboring *Methylomonas* sp. DH-1 to NiCl$_2$, wild-type cells and the cells containing the *tetA* gene in the plasmid (*E. coli*) or in the genome (*Methylomonas* sp. DH-1) were incubated in an LB (*E. coli*) or NMS (*Methylomonas* sp. DH-1) agar plate with different concentrations of NiCl$_2$ (0.5–8 mM). The optimal NiCl$_2$ concentrations that were determined to counter select *tetA*-removed cells were 2 mM and 4 mM for *E. coli* and *Methylomonas* sp. DH-1, respectively.

**Plasmid removal assay in *E. coli* using CPP-conjugated I-SceI**. For the plasmid removal experiment, three *E. coli* strains carrying a target plasmid with a different replication origin were prepared. The plasmids contained an I-SceI restriction site and a replication origin (pSC101, ColE1, or RSF1030). Cells at a density of 5 × 10$^7$ cells/50 μL were treated with 50 μL of CPP-conjugated I-SceI (0.05 μg/μL) and incubated with 20 mM Tris-HCl (at pH 7.5) for 3 h after electroporation. Then, the

buffer was replaced with 1 × CutSmart buffer, because I-SceI requires the Mg$^{2+}$ included in the CutSmart buffer[64] as a cofactor. The cells were further incubated at 37 °C for 2 h and then spread on LB agar plates containing 2 mM NiCl$_2$. The colonies grown on NiCl$_2$-containing plates were transferred to an LB agar plate and a tetracycline-containing LB agar plate in order to confirm whether the plasmids were completely removed from the cells.

**Marker gene excision in *Methylomonas* sp. DH-1 using CPP-conjugated Cre recombinase**. For the marker gene excision experiment, the pIns-LKTL plasmid was transformed into *Methylomonas* sp. DH-1, and then the marker genes flanked with loxP sites (loxP-*kan*$^R$-*tetA*-loxP) were integrated into a non-coding region of the *Methylomonas* sp. DH-1 genome. To select correctly knocked-in cells, the transformed cells were grown on an NMS agar plate containing kanamycin (10 μg/mL) and 30% methane for 3–7 days. The knocked-in cells were harvested from the plate and washed twice with distilled water, and 50 μL of cell suspension was mixed with 50 μL of CPP-conjugated Cre recombinase (0.05 μg/μL). The mixture was transferred to a pre-chilled electroporation cuvette with a 0.2 cm gap, and then electroporation was performed using an electroporator (MicroPulser, Bio-Rad) at the preprogrammed Ec2 setting. After electroporation, cells were incubated in two types of recovery buffers (20 mM Tris-HCl at pH 7.5 or NMS medium) for 3 h and then spread on NMS agar plates containing 4 mM NiCl$_2$. The colonies grown on the NiCl$_2$ plates were transferred to an NMS agar plate and a kanamycin-containing NMS agar plate to confirm the excision of the marker genes.

**Tests for cytotoxicity and penetration efficiency of CPPs in mammalian cells**. HEK293 and Chinese Hamster Ovary (CHO) cells were used as a test platform for mammalian systems. The cells were cultured in Dulbecco's Modified Eagle Medium containing 10% fetal bovine serum and 1 × penicillin/streptomycin (all from Invitrogen, Waltham, MA, USA). For cytotoxicity test, 3 × 10$^4$ cells/cm$^2$ were plated on wells of 0.1% gelatin-coated (only for HEK293) 96-well plate. One day later, the cells were exposed to 5 μg of CPPs for 24 h. After washing with 1 × PBS twice, the cells were subjected to the viability test using a Chromo-CK™ assay kit (Monobio, Seoul, Korea) according to manufacturer's instruction. For penetration efficiency test, cells were plated on 96-well plates at a density of 5 × 10$^4$ cells/cm$^2$, and then they were cultured with 5 μg of CPPs for 16 h. After washing with 1 × PBS twice, the fluorescence emitting from the cells was measured with a multi-detection microplate reader (Hidex, Turku, Finland) at 575 nm. The cells treated with DMSO were used as a control in both tests. Each test was performed in biological triplicates, and data were presented as mean ± standard error.

**Statistics and reproducibility**. Data obtained from at least three independent experiments were analyzed using GraphPad Prism v7.0 (GraphPad Software, Inc.). Replicates were plotted using the average and standard error of the mean (SEM). As a significance test, a *t* test was used with a cutoff *p* value of 0.05 or 0.001.

**Reporting summary**. Further information on research design is available in the Nature Research Reporting Summary linked to this article.

## Data availability

All data generated or analyzed during this study are included in Supplementary Data 1 and 2. Three representative plasmids constructed in this study (CPP 70-GFP, CPP 70-I-SceI, and CPP 70-Cre recombinase) are available at Addgene and their Addgene IDs are 165040, 165041, and 165042, respectively. Any remaining information can be obtained from the corresponding author upon reasonable request.

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

## Acknowledgements

This research was supported by the C1 Gas Refinery Program through the National Research Foundation of Korea (NRF) funded by the Ministry of Science,

ICT and Future Planning (NRF-2016M3D3A1A01913244). This work was also supported by the NRF grant funded by the Korean government (MSIT) (No. NRF-2018R1A5A1025077).

## Author contributions

H.L., J.R., and D.N. contributed to idea generation, methods, and analysis. H.L., J.R., K.M.T., H.K., K.E.L., Y.O., and M.C. conducted the experiments and interpreted the data. D.N. supervised the work. B.J., W.P., and D.K. conducted the mammalian cell experiments. H.L. and D.N. wrote the paper, which was edited and approved by the other coauthors.

## Competing interests

The authors declare no competing interests.
