## [Peer Review File · Communications Biology]

Reviewers' comments:

Reviewer #1 (Remarks to the Author):

In this study, the authors investigated the utilization of cell-penetrating peptides (CPPs) in genetic engineering of prokaryotic cells, *E. coli* and *Methylomonas sp, DH-1*. 98 CPPs were examined for their *E. coli* cell penetration efficiency and toxicity. Furthermore, a computational penetration efficiency prediction model was developed, which could help design new CPPs with good efficiency. Two CPPs with high penetration efficiency when conjugated with GFP protein were applied to restriction enzyme-dependent plasmid removal in *E. coli* and Cre-mediated marker removal in *Methylomonas sp, DH-1*. The experiments were carefully conducted and well described. Although CPPs have been extensively studied in mammalian cells, this study might be meaningful in providing information on the utilization of CPPs in protein delivery in prokaryotic cells. The following is some comments on this paper.

1. Considering the fact that plenty of information is available for CPPs studied in mammalian cells, it would be useful to compare the efficiency and toxicity of CPPs in prokaryotes and mammalian cells for the future usage of CPPs in prokaryotes. It might not be necessary to test all 98 CPPs, but testing a selected set of representative CPPs (high and low efficiency, high and low toxicity) would be informative to understand the similarity and difference when applying CPPs in different cell types. At least, the efficiencies of different CPPs could be compared based on the information available from previous publications. Since plasma membrane structures are largely conserved between prokaryotes and eukaryotes, such study might allow to predict the contribution of other characteristics of the gram-negative bacterial cell envelope on the efficiency or toxicity of CPP.
2. P22, lines 391-392 'due to the absence of episomal plasmids in *Methylomonas sp. DH-1*': *Methylomonas sp. DH-1* does have natural episomal plasmids, but genetic engineering tools using episomal plasmids are not currently available. So, please correct the sentence to prevent misleading.

Reviewer #2 (Remarks to the Author):

In the manuscript by Lee et al., a CPP library was constructed and a set of CPPs were identified that could penetrate bacterial cells with minimum or no impact on cell viability. For the identified top CPP candidates, their abilities to deliver macromolecule to bacteria were evaluated using a conjugated GFP protein. The authors then applied these CPPs to deliver I-SceI and Cre recombinase proteins to bacteria as proof-of-concept studies for potential applications. Overall, this study is well designed and performed and may have important implications to bacterial engineering. Therefore, I would recommend the publication of this manuscript in *Communications Biology*. However, the following points should be addressed before the acceptance for publication.

Major

1. The authors observed that fusion of certain CPPs to I-SceI could disrupt the folding of cargo proteins, as shown in SI Fig. 5. It is thus necessary to further discuss this observation. For example, additional text can be added to Pg. 25 Line 455 to precaution the potential folding issue when CPPs are fused to proteins.

Minor

1. It appeared that the authors established/optimized an electroporation-based CPP screening system in the first section of Results. The authors may want to clearly indicate this, either in the sub-section title or in the beginning paragraph, so that the readers can better understand the purpose of these experiments.
2. The authors may want to check the figure numbering in the text. For example, in the section "CPP penetration efficiency evaluation", the authors jumped from Fig. 3a to Fig. 3d. Either the

figure or figure numbering need to be re-organized.

Point-by-point responses to the reviewers:

First of all, we would like to appreciate the reviewers for their encouraging and helpful comments to improve our manuscript. We hope that by addressing these comments we are able to create a considerably stronger manuscript that will be suitable for publication and also appeal to the readers of *Communications Biology*. Please find our point-by-point responses below.

● Reviewer 1:

General response: We appreciate this reviewer for the encouraging words regarding our work: "The experiments were carefully conducted and well described. Although CPPs have been extensively studied in mammalian cells, this study might be meaningful in providing information on the utilization of CPPs in protein delivery in prokaryotic cells."

Point 1. Considering the fact that plenty of information is available for CPPs studied in mammalian cells, it would be useful to compare the efficiency and toxicity of CPPs in prokaryotes and mammalian cells for the future usage of CPPs in prokaryotes. It might not be necessary to test all 98 CPPs, but testing a selected set of representative CPPs (high and low efficiency, high and low toxicity) would be informative to understand the similarity and difference when applying CPPs in different cell types. At least, the efficiencies of different CPPs could be compared based on the information available from previous publications. Since plasma membrane structures are largely conserved between prokaryotes and eukaryotes, such study might allow to predict the contribution of other characteristics of the gram-negative bacterial cell envelope on the efficiency or toxicity of CPP.

Response: Thank you for the valuable comment. As suggested, we measured penetration efficiency and toxicity of the selected CPPs in two different mammalian cells, CHO and HEK293 cells, which are commonly used in laboratories. For the investigation, we selected 10 highly efficient and 10 highly inefficient CPPs, and 10 highly toxic and 10 highly safe CPPs. The results are shown in Fig. 8, and a new paragraph was added to the main text. The selected CPPs showed very different properties from cell type to type, which may result from the differences of the cells (human kidney cell, hamster ovary cell, and *E. coli*).

Fig. 8. Penetration efficiencies and cytotoxic effects of the CPPs in HEK293 and CHO cells, and comparison of CPP effects in different cell types

a, b Ten highly efficient CPPs and ten highly inefficient CPPs in *E. coli* were selected, and their cell penetration efficiency was measured in HEK293 and CHO cells. **c-e** Linear correlations of the penetration efficiencies among the three cell types. **f, g** Ten highly toxic CPPs and ten highly safe CPPs in *E. coli* were selected, and their cytotoxicity was measured in HEK293 and CHO cells. **h-j** Linear correlations of the cytotoxicity among the three cell types.

(page 15)

CPP penetration efficiency and toxicity in mammalian cells (HEK293 and CHO cells)

To date, mostly CPPs have been studied and utilized in mammalian cells³⁶, but in this study we studied CPPs in bacteria, *E. coli*. Thus, it would be very useful to find the similarities and differences of CPPs between prokaryotes and eukaryotes. For example, the CPPs derived from the HIV Tat sequence work well in both *E. coli* and mammalian cells^{36,57}, while CPP 63 (KLPVM) was known as a highly efficient CPP in mammalian cells^{36,58}, but showed low efficiency in *E. coli* (Fig. 3a).

For comparison, we selected ten highly efficient CPPs and ten highly inefficient CPPs from our study and evaluated their efficiencies in two different mammalian cells, HEK293 and Chinese Hamster Ovary (CHO) cells. As shown in Fig. 8a, generally the ten efficient CPPs in *E. coli* also showed high efficiency in HEK293 cells. Conversely, the efficient CPPs in *E. coli* showed low penetration efficiencies in CHO cells (Fig. 8b). Surprisingly, when their correlations were analyzed, we could not find any statistical correlations of the penetration efficiency among the three cells (Fig. 8c-e).

For toxicity comparison, we selected ten highly toxic CPPs and ten highly safe CPPs, and measured their toxicities in HEK293 and CHO cells. Most of the CPPs did not show significant toxicity in both HEK293 and CHO cells, but CHO cells were more resistant against CPP toxicity than HEK293 cells (Fig. 8f and g). When correlations were analyzed, we also could not find any correlations among the three cells (Fig. 8h-j).

The low correlations represent that CPP penetration efficiency may be different from cells to cells, and this may hinder the development of universal CPPs that are efficient in many different cell types. However, the three cells used in this study are very different: prokaryotic cell (*E. coli*), human embryonic kidney cell (HEK293), and hamster ovary cell (CHO). Thus, more studies are required in order to find the common and distinct features of CPPs.

(page 18)

Despite of the wide applicability of CPPs in biotechnological engineering, CPPs may show different

properties (penetration efficiency and toxicity) from cell types to types: one efficient CPP developed in one cell type may not work well in other cell types. This may require iterative screening for efficient CPPs in every cell type. However, since we used only three very distinct cell types (prokaryote, human kidney cell, and hamster ovary cell), more studies are required to draw a concrete conclusion.

(page 28)

Tests for cytotoxicity and penetration efficiency of CPPs in mammalian cells

HEK293 and Chinese Hamster Ovary (CHO) cells were used as a test platform for mammalian systems. The cells were cultured in Dulbecco's Modified Eagle Medium containing 10 % fetal bovine serum and 1× penicillin/streptomycin (all from Invitrogen, Waltham, MA, USA). For cytotoxicity test, 3×10^4 cells/cm² were plated on wells of 0.1 % gelatin-coated (only for HEK293) 96-well plate. One day later, the cells were exposed to 5 µg of CPPs for 24 hours. After washing with 1× PBS twice, the cells were subjected to the viability test using a Chromo-CK™ assay kit (Monobio, Seoul, Korea) according to manufacturer's instruction. For penetration efficiency test, cells were plated on 96-well plates at a density of 5×10^4 cells/cm², and then they were cultured with 5 µg of CPPs for 16 hours. After washing with 1× PBS twice, the fluorescence emitting from the cells was measured with a multi-detection microplate reader (Hidex, Turku, Finland) at 575 nm. The cells treated with DMSO were used as a control in both tests. Each test was performed in biological triplicates, and data was presented as mean ± standard error.

Point 2. P22, lines 391-392 'due to the absence of episomal plasmids in *Methylomonas sp. DH-1*': *Methylomonas sp. DH-1* does have natural episomal plasmids, but genetic engineering tools using episomal plasmids are not currently available. So, please correct the sentence to prevent misleading.

Response: As commented, *Methylomonas sp. DH-1* does have a natural episomal plasmid, but episomal plasmids as a tool for genetic engineering are not available yet. Thus, to avoid confusions, we clarified this in the manuscript as below.

(page 5)

However, the metabolic engineering of this methanotroph is challenging due to the lack of genetic manipulation tools, including the absence of **artificial plasmids for genetic engineering**.

(page 14)

However, **since currently there are no artificial plasmids for the genetic engineering of *Methylomonas sp. DH-1*, it is difficult to remove** the marker gene. Thus, a protein delivery method is required for marker excision.

● **Reviewer2:**

General response: We appreciate this reviewer for the encouraging words regarding our work: "Overall, this study is well designed and performed and may have important implications to bacterial engineering."

Point 1. The authors observed that fusion of certain CPPs to I-SceI could disrupt the folding of cargo proteins, as shown in SI Fig. 5. It is thus necessary to further discuss this observation. For example, additional text can be added to Pg. 25 Line 455 to precaution the potential folding issue when CPPs are fused to proteins.

Response: We agreed that there could be a potential folding issue when a CPP is fused to a protein cargo, probably due to the charged amino acids. Thus, this issue was further discussed in the discussion section of manuscript as below.

(page 17)

However, there could be a potential folding issue when a CPP is fused to a protein, since CPPs are generally are highly charged. For example, the purified CPP-conjugated I-SceI showed no activity without chaperone expression, while purified I-SceI showed normal restriction activity. Therefore, it should be considered whether CPP interferes with the folding of the cargo protein or not.

Point 2. It appeared that the authors established/optimized an electroporation-based CPP screening system in the first section of Results. The authors may want to clearly indicate this, either in the subsection title or in the beginning paragraph, so that the readers can better understand the purpose of these experiments.

Response: As recommended, we have added sentences in the beginning of the paragraph as below, and we think this will help the readers understand the purpose of the electroporation experiments more clearly and better.

(page 5)

Improved delivery efficiency of CPP conjugates by electroporation in *E. coli*

In bacteria, CPPs have been used without additional treatments, but recently, the chemical treatment of cells has improved the delivery efficiency of CPP conjugates⁴⁴. Electroporation has been widely used as an effective method for biomacromolecule delivery, and electroporation is generally more efficient than chemical treatments, such as that with CaCl₂⁴⁹. **In this study, we developed and optimized a new electroporation-based delivery method to further improve CPP delivery efficiency, and used the method for CPP delivery experiments.**

Point 3. The authors may want to check the figure numbering in the text. For example, in the section “CPP penetration efficiency evaluation”, the authors jumped from Fig. 3a to Fig. 3d. Either the figure or figure numbering need to be re-organized.

Response: As commented, we have corrected them.

REVIEWERS' COMMENTS:

Reviewer #1 (Remarks to the Author):

The revised manuscript was significantly improved and acceptable for publication.

Reviewer #2 (Remarks to the Author):

All my concerns have been adequately addressed. Thus I recommend the publication of this work in Commun Biol.